# Intravenous Immunoglobulin-Induced Aseptic Meningitis—A Narrative Review of the Diagnostic Process, Pathogenesis, Preventative Measures and Treatment

**DOI:** 10.3390/jcm11133571

**Published:** 2022-06-21

**Authors:** Anna Kretowska-Grunwald, Maryna Krawczuk-Rybak, Malgorzata Sawicka-Zukowska

**Affiliations:** Department of Pediatric Oncology and Hematology, Medical University of Bialystok, Jerzego Waszyngtona 17, 15-274 Bialystok, Poland; rybak@umb.edu.pl (M.K.-R.); malgorzata.sawicka-zukowska@umb.edu.pl (M.S.-Z.)

**Keywords:** aseptic meningitis, intravenous immunoglobulins, drug-induced meningitis

## Abstract

Intravenous immunoglobulins (IVIGs) are widely used in the treatment of numerous diseases in both adult and pediatric populations. Higher doses of IVIGs usually serve as an immunomodulatory factor, common in therapy of children with immune thrombocytopenic purpura. Considering the broad range of IgG applications, the incidence of side effects in the course of treatment is inevitable. Aseptic meningitis, an uncommon but significant adverse reaction of IVIG therapy, can prove a diagnostic obstacle. As of April 2022, forty-four cases of intravenous immunoglobulin-induced aseptic meningitis have been reported in the English-language literature. This review aims to provide a thorough overview of the diagnostic process, pathophysiology, possible preventative measures and adequate treatment of IVIG-induced aseptic meningitis.

## 1. Introduction

### 1.1. Intravenous Immunoglobulins

Intravenous immunoglobulins (IVIGs) are a blood product derived from the serum of numerous human donors, mainly composed of proteins (mostly IgG (>95%), IgA, IgM and albumins) and supplementary additives-sugar, salts, solvents, and detergents [1]. Assuring low pH level of the products is beneficial to preventing aggregation and transmission of viral infections [2]. In order to ensure product safety, adequate manufacturing guidelines should be met [3]. Intravenous immunoglobulins have been approved for treatment of numerous diseases, differing in dosage. Lower doses (in total of 200–400 mg/kg) are usually used as a “replacement” in patients with antibody deficiencies. Higher doses of IVIGs (in total of 2 g/kg) usually serve as an immunomodulatory factor, initially used in children with immune thrombocytopenic purpura (ITP) [4,5]. Nowadays, besides ITP, higher levels of IVIGs are also being administered to patients treated for various autoimmune and inflammatory diseases, including Guillain–Barre syndrome, chronic inflammatory demyelinating polyneuropathy, Kawasaki disease, myasthenia gravis and systemic lupus erythematosus as well as in primary or acquired immunodeficiency diseases [4,6]. Recent studies also describe the promising outcome of immunoglobulin therapy in patients diagnosed with COVID-19 [7].

### 1.2. Adverse Reactions of IVIG

Considering the wide spectrum of IVIG therapy applications, there is a certainty of side effect occurrence. As many as 40% of intravenous IgG infusions result in the incidence of adverse reactions, most being mild, uncomfortable but not very life-threatening [8]. The occurrence of adverse reactions associated with intravenous immunoglobulin administration is quite ambiguous, ranging from 1 to 81% depending on the literature, the most frequent IVIG adverse events being fever and headache [8,9,10,11]. Most of the symptoms are not considered to be severe-occurring in the first hour of treatment and disappearing within several hours [12]. Delayed adverse reactions (6 h to 1 week after infusion) although uncommon are far more serious. Those which can prove life-threatening account for less than 5% of cases and include—hemolytic anemia, encephalopathy, anaphylaxis, renal complications, thrombosis/embolism, and colitis [12,13,14,15]. Skin reactions are mostly associated with subcutaneous immunoglobulin infusions [12,16,17,18]. True anaphylactic reactions occur rarely and are thought to be associated with innate IgA deficiency [19].

### 1.3. Aseptic Meningitis

Aseptic meningitis (AM) is considered an uncommon adverse reaction associated with the course of intravenous immunoglobulin therapy. It can be linked to 0.067% of all infusions with the incidence being higher following IVIG treatment in patients with Guillain-Barre syndrome [20]. It usually accounts for a delayed reaction (after 6 h) of a moderate severity, with most AM cases having the tendency to occur within 48 h of infusion [12]. A history of migraine headaches is thought to increase susceptibility of IVIG-treated patients to this side effect, regardless of the immunoglobulin brand used or the rate of the infusion [21,22,23]. Although aseptic meningitis can occur in patients receiving immunoglobulins regardless of the dosage, it is more likely to be associated with those treated with a higher dose (1–2 g/kg) of immunoglobulins (thrombocytopenia, Kawasaki disease) [13]. Besides IVIG-related, very few other associations in the literature have been made between the diagnosis of ITP and the incidence of aseptic meningitis, showing a direct cause-and-effect relationship between intravenous immunoglobulin treatment and occurrence of aseptic meningitis. A study by Mohammed A. Aldriweesh et al. identifies Varicella Zoster Virus as both the most common viral infection among patients with ITP and one of the leading viral causes of aseptic meningitis [24]. The possibility of a concomitant incidence of both ITP and AM was also reported in the context of a potential adverse reaction of the measles-mumps containing vaccination by Silvia Perez-Vilar et al. [25]. On the contrary, patients with Kawasaki disease may manifest in other, less common ways such as aseptic meningitis. Therefore, the incidence of AM in the course of Kawasaki disease therapy may not always be directly associated with intravenous immunoglobulin infusion [26,27,28].

### 1.4. DIAM—Drug-Induced Aseptic Meningitis

There have been numerous reports published in the English-language literature describing the causative agents of drug-induced aseptic meningitis (DIAM). An investigation of nearly 300 patient cases performed by Bihan et al. suggests immunoglobulins (IVIGs) to be the most common cause of DIAM [29]. Noteworthy, a literature review by Jolles and Hopkins identifies non-steroidal anti-inflammatory drugs (NSAIDs) as such [14]. Antimicrobials-cotrimoxazole and cephalosporins of various generations are also commonly associated with drug-induced aseptic meningitis [30]. Intrathecally administered drugs such as methylprednisolone and anesthetics can be a direct cause of meningeal irritation [31].

Despite numerous publications on intravenous immunoglobulin-induced adverse reactions, not much is to date reported about the incidence, pathogenesis, treatment, and prevalence of IVIG-associated acute meningitis.

## 2. Materials and Methods

A comprehensive search was conducted last in April 2022 in PubMed online electronic database using the key phrases (“immunoglobulin” OR “IVIG”) AND “aseptic meningitis”. No timeframe restrictions were assigned for the selected publications. The articles were selected following the PRISMA guidelines. A total of 284 articles were found. Before the initial screening, five articles were found to have duplicates, therefore they were removed. Additionally, 31 papers were unable to be screened (i.e., no abstract available). A total of 248 publications were screened and 22 non-English articles were excluded. Five articles were unable to be retrieved. Out to the 221 reports, which were assessed for eligibility, 21 were rejected as they involved non-human subjects. A total of 128 articles were identified after thorough analysis as irrelevant to the review. Finally, a total of 72 papers were analyzed in this article. All authors conducted the search separately in order to provide the most accurate results and avoid the risk of bias. In Figure 1, we present the schematic diagram of the selection process of articles chosen for this review.

## 3. Characteristics of IVIG-Induced Aseptic Meningitis Reported Cases

As of April 2022, to our knowledge, forty-four cases of intravenous immunoglobulin-associated aseptic meningitis have been reported in the English-language literature. The median age of the patients was 22.4 years, with twenty-four (54.50%) of the published cases involving children fourteen and younger. Therefore, the incidence of this adverse reaction should be especially brought to the attention of pediatricians. IVIG-induced aseptic meningitis was diagnosed most often during the course of treatment of immune thrombocytopenic purpura (21 patients) and Kawasaki disease (5 patients) (Table 1). Noteworthy, roughly around 45% of the reported patient cases (20 patients) were treated with empiric antibiotic therapy regardless of the cerebrospinal fluid (CSF) analysis. 50% of patients with lymphocyte predominance in the CSF and only 46.80% of those with the majority of granulocytes in the CSF received antibiotic treatment (Table 1 and Table 2). The onset of aseptic meningitis symptoms greatly varied depending on the patient, with the earliest side effects reported within 24-h of the first infusion to even 10 days after the last one [20,22]. Interestingly, white blood cell count in the cerebrospinal fluid of the described patients showed a broad range from 0.0000018 × 10^9^ L in a patient with aseptic meningitis in the course of treatment of Guillain-Barre syndrome to 7.44 × 10^9^ L in a child with ITP [33,34]. The gender distribution in the reported cases points to a slight dominance of males (22 males, 21 females, 1 not stated). Published literature on the subject of IVIG-induced aseptic meningitis refers to intravenous immunoglobulins under thirteen different brand names (Table 2).

## 4. Diagnosis of Aseptic Meningitis in Clinical Practice

Aseptic meningitis, a diagnosis of exclusion, refers to a process free from bacterial, viral, or fungal contamination [55]. Aseptic meningitis can prove a diagnostic obstacle as both clinical and cerebrospinal fluid (CSF) findings can make it indistinguishable from infectious meningitis [52,56]. Computed tomography (CT) of the brain is generally free of acute changes [50]. The most common symptom of aseptic meningitis usually includes headache of various clinical presentations from intermittent to constant, global to localized [57]. Nausea, vomiting, photophobia, and fever can also occur [36,38,53]. Neurological examination might indicate nuchal rigidity, positive Kernig’s sign but without any focal neurological signs [37,58]. Peripheral blood examination usually reveals leukocytosis [41]. There is no evidence of reported deaths due to aseptic meningitis in the course of IVIG-therapy [19].

The analysis of the cerebrospinal fluid is a crucial step in determining the correct diagnosis. Viral (cytomegalovirus, enterovirus, Herpes simplex virus, Varicella zoster virus) and bacterial etiology (Neisseria meningitidis, Streptococcus pneumoniae, Cryptococcus neoformans, Listeria monocytogenes) of meningitis should be excluded [52]. Numerous examinations of CSF of patients with suspected drug-induced aseptic meningitis revealed pleocytosis of hundred to several thousand cells with neutrophil dominance [53,55], leukocytes level in the CSF reaching as high as 7440 × 10^6^ g/dL [33]. On the contrary Jain et al. reported a case of a patient diagnosed with Guillain-Barre syndrome, who developed IVIG-induced aseptic meningitis manifesting with lymphocyte-predominant (85%) CSF [59]. Elevated eosinophiles could also be observed [30,34]. Protein levels in the CSF seem to be increased with a glucose level decreased or within the normal range [31]. Sekul et al. suggested that an allergic hypersensitivity reaction can be justified by the presence of eosinophilia in CSF [22].

According to a study done by Kattamis et al., there seems to be a higher incidence of IVIG-associated neurological adverse reactions in older children. Whether this is a result of the fact that children, especially infants, cannot complain of subjective IVIG-AEs such as headache, nausea, and abdominal pain is yet to be determined [9] There have been several studies describing cases of children, who were old enough (6, 9 and 10 y/o) to precisely locate their complaints [33,47]. Nevertheless, it is noteworthy to closely monitor children with both personal and familiar history of headaches and autoimmune diseases, including sarcoidosis, systemic lupus erythematosus granulomatosis with polyangiitis, as these patients are especially prone to developing aseptic meningitis after treatment with high doses of intravenous immunoglobulins [31,55].

## 5. Pathophysiology of IVIG-Associated Aseptic Meningitis

Although the pathophysiology of IVIG-induced aseptic meningitis is to date ambiguous, there are several possible mechanisms, which are thought to explain the incidence of this adverse reaction. These include the hypersensitivity especially immune complex-mediated (type III) and cell-mediated (type IV) of the leptomeninx, direct meningeal irritation by the drug, the effect of the release of inflammatory cytokines due to the interactions of IgG and vessel antigens in the meninges [6,9,53]. The presence of eosinophils in the CSF and the relatively rapid onset of symptoms (usually below 48 h) after immunoglobulin infusion strongly suggest ethology related to hypersensitivity [60]. In search for more thorough understanding of the pathogenesis of aseptic meningitis, Asano et al. investigated the presence of several cytokines and chemokines in the CSF of patients with both aseptic and viral meningitis. The study revealed that the level of monocyte chemoattractant protein-1 (MCP-1) was much higher in the CSF of patients diagnosed with IVIG-induced meningitis. The increase of MCP-1 in the CSF potentially results in the activation of monocytes and development of aseptic meningitis [61].

Despite several studies suggesting no apparent relationship between the incidence of adverse reactions and the brand of the immunoglobulins used [30], some authors imply this association [62] with Jarius et al. pointing to antineutrophil antibodies present only in some IVIGs [12,59]. Even though, there have been several literature reports concerning adverse events related to sugar additives of the intravenous immunoglobulin products, they mostly focus on renal insufficiency [63].

Aseptic meningitis is usually related to high dose IVIG administration [30]. Interestingly, according to St-Amour et al. roughly 0.01% of systematically injected immunoglobulins cross the blood brain barrier and can be detected mostly in the microvessels [64]. Hipervicosity of the blood, a result of IVIG-induced increase of serum total protein, seems to also play a role in the pathogenesis of adverse reactions related to intravenous immunoglobulin administration. Therefore, high serum total protein could possibly be considered a predictive risk factor for the possible occurrence of IVIG-related neurological side effects [9].

## 6. Prevention of Aseptic Meningitis

One large 200 patient group study analyzed by Katz U et al. considered slow infusion rate and good hydration measures of preventing aseptic meningitis [Figure 2]. These precautionary methods are additionally thought to prevent thrombosis and renal failure [13]. Noteworthy, blood pressure and urine production should be monitored while the drug is being administered [65]. The infusion should not be faster than 6 g/h and at a dilution of 3%. Kareva et al. suggest that the infusion rate is directly proportional to the severity of adverse events [66]. Although slow infusion rate and lower initial dosage of IVIGs [67] are known detrimental factors of adverse reaction prevention caused by IVIG therapy, they do not seem to always exclude their incidence [9]. Some also recommend paracetamol with additional codeine as a possible preventative measure [30]. Pretreatment with oral or intravenous corticosteroids prior to high-dose immunoglobulin infusion is a dividing matter although it is widely used as routine practice [9,13]. Steihm et al. suggest that premedication with steroids should only be limited to those patients with adverse reactions due to IVIG in their medical history or are thought to be at risk of developing an unwanted complication [12]. Therefore, obtaining a thorough medical history is crucial for undertaking specific precautionary measures [56]. Patients diagnosed with renal failure, diabetes mellitus, or hypertension should not receive IVIG products containing sucrose [68]. According to Jolles et al. corticosteroids do not fully prevent adverse outcomes [30]. Although this method of premedication might be considered debatable, a study by Carcao et al. indicated that combined therapy of intravenous methylprednisolone and intravenous immunoglobulins seems to raise platelet levels faster than IVIG therapy alone in patients with immune thrombocytopenia [69]. Considering that hypersensitivity reaction is one of the possible causes of IVIG-related aseptic meningitis, antihistamine drugs such as diphenhydramine are also recommended to be used as premedication. Furthermore, their combination with hydrocortisone should be considered a possibility [62].

## 7. Treatment of IVIG-Induced Aseptic Meningitis

Considering the fact that aseptic meningitis is mostly a diagnosis of exclusion [30], it is crucial to rule out an infectious background of the presenting symptoms. The symptoms usually alleviate shortly after immediate cessation of the causative agent and 2–3 days of symptomatic treatment [70]. On occasion empiric antimicrobial therapy preferably with ceftriaxone is instantly initiated as a precautionary measure and is discontinued after negative blood cultures [41,52]. Treatment of aseptic meningitis relies mainly on symptom alleviation i.e., antiemetics in case of nausea and analgesics for the headache [18]. High fluid intake is not to be discouraged. Given that several studies report the presence of eosinophils in the CSF of patients with IVIG-induced aseptic meningitis, cetirizine or other second-generation antihistamine drugs are also considered an additional line of treatment [30]. If signs of aseptic meningitis appear during an IVIG infusion, treatment should be stopped immediately. Symptoms should alleviate within 48 h after cessation [34]. Noteworthy, the literature describes a single case report, where the infusion was continued despite occurrence of symptoms suggesting aseptic meningitis. The clinical presentation did not differ from the time the drug administration was ceased [37]. In a case described by Jain et al. IVIG therapy was reintroduced at a slower infusion rate without aseptic meningitis reoccurrence [59]. Changing the IVIG product brand could also be beneficial in preventing side effects [46]. No therapeutic efficacy seems to be lost when choosing subcutaneous immunoglobulins as an alternative treatment method [71,72].

## 8. Conclusions

Aseptic meningitis, although uncommon, should always be taken into consideration by physicians as a possible delayed adverse reaction of intravenous immunoglobulin infusion. The clinical manifestation is often ambiguous causing the diagnosis to prove a diagnostic obstacle. Analysis of the CSF is a crucial approach in determining the correct diagnosis, resulting in the introduction of adequate treatment. Previous medical history, frequent occurrence of headaches, and family history of autoimmunological diseases all require taking precautionary measures in the form of premedication in order to prevent the incidence of aseptic meningitis. Treatment relies mainly on the cessation of the causative agent with additional symptomatic therapy.

## Figures and Tables

**Figure 1 jcm-11-03571-f001:**
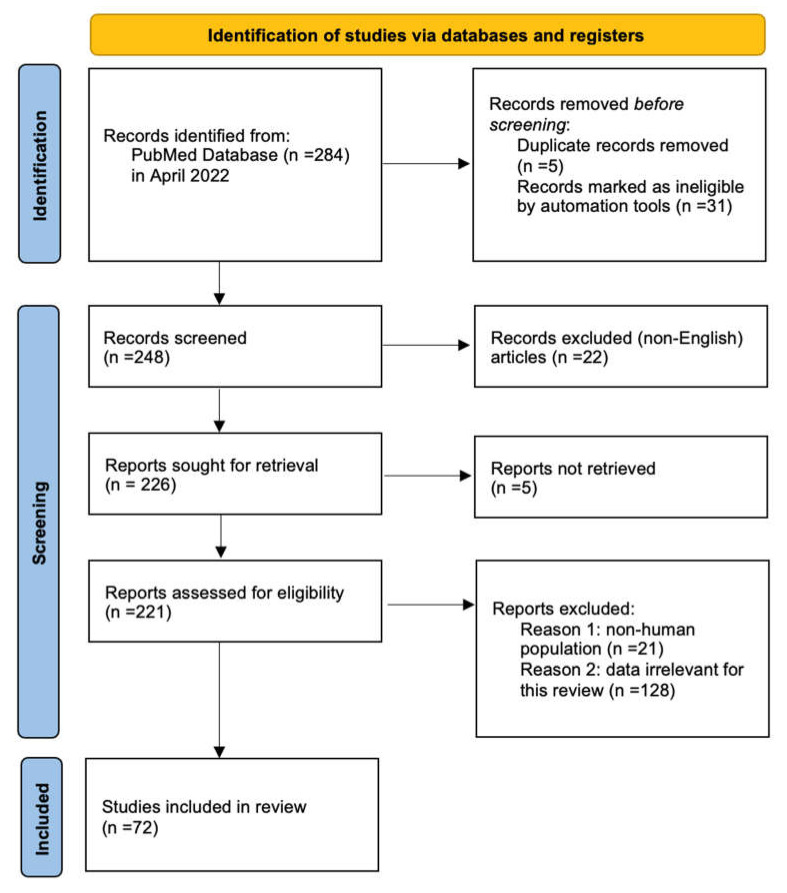
PRISMA flow diagram 2020. Adapted from Page et al. [32].

**Figure 2 jcm-11-03571-f002:**
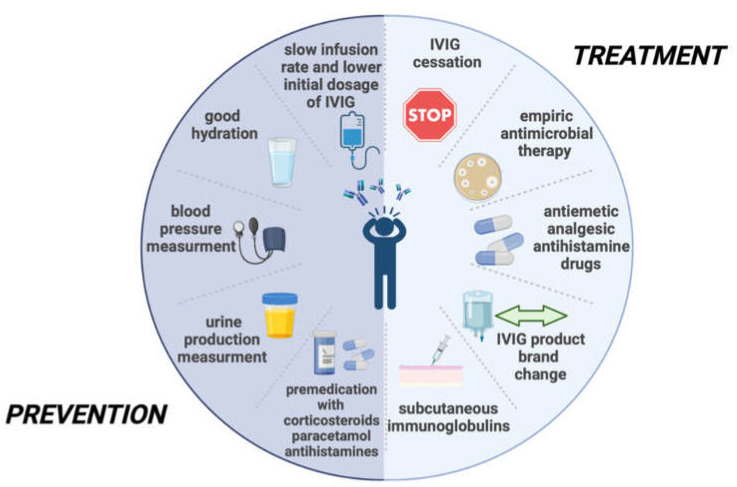
Treatment and preventative measures of IVIG-induced aseptic meningitis [18,30,41,50,52,62,65,66,67,70,71]. This figure was created with BioRender.com (https://biorender.com/ (accessed on 12 May 2022).

**Table 1 jcm-11-03571-t001:** Cases of documented IVIG-related aseptic meningitis in the years 1981–2019.

Diagnosis	Sex	Age	IVIG Dose	Brand	Onset	WBC × 10^9^ L csf	CSF Cytosis	Treatment	Source
ITP	M	6	1 g/kg	NS	10 h after the last dose	0.35	94% granulocytes	Cefuroxime iv for 3 days	[35]
ITP	M	9	0.4 g/kg	Sandoglobulin	12 h after the last dose	2.45	98% granulocytes	Prednisone 3 mg/kg for 4 days	[36]
ITP	M	4	0.4 g/kg	Sandoglobulin	2nd day	2.0	67% granulocytes	Self-limiting	[37]
ITP	M	4	0.4 g/kg	Globuman	2nd day	NS	NS	Self-limiting	[37]
ITP	F	25	1 g/kg	Intragam	Evening of the 3rd day	0.022	64% lymphocytes	Analgetic	[18]
ITP	F	26	0.4 g/kg	Intragam	3rd day	0.131	72% granulocytes	Ampicillin	[18]
Recalcitrant pemphigus vulgaris	F	26	2 g/kg	NS	3rd day	0.08	NS	Self-limiting	[38]
ITP	F	14	0.4 g/kg	NS	2 days after last infusion	0.14	70% granulocytes	Floctafenine	[39]
ITP	M	7	0.4 g/kg	NS	3rd day	NS	Ns	NS	[40]
Polymyositis	M	40	2 g/kg	NS	Within 24 h after infusion	0.75	87% granulocytes	Narcotic analgesics and antiemetic agents	[22]
Dystrophy	M	7	2 g/kg	NS	Within 24 h after infusion	0.22	85% granulocytes	Narcotic analgesics and antiemetic agents	[22]
Multifocal motor neuropathy with conduction block	M	37	2 g/kg	NS	Within 24 h after infusion	1.17	85% granulocytes	Narcotic analgesics and antiemetic agents	[22]
Paraproteinemic polyneuropathy	F	61	2 g/kg	NS	Within 24 h after infusion	0.016	58% lymphocytes	Narcotic analgesics and antiemetic agents	[22]
Dermatomyositis	F	48	2 g/kg	NS	Within 24 h after infusion	0.001	92% lymphocytes	Narcotic analgesics and antiemetic agents	[22]
ITP	F	27	0.4 g/kg	Sandoglobulin	3rd day	3.26	97% granulocytes	Ceftriaxone 2 g iv for 2x/d for 5 days, steroids	[41]
ITP	F	2	0.4 g/kg	Prepared with polyethylene glycol	7 days after therapy	0.451	9% granulocytes	Self-limiting	[42]
ITP	M	7	0.4 g/kg	Sandoglobulin	1 h after 2nd dose	2.45	88% granulocytes	Self-limiting	[43]
ITP	M	10	0.4 g/kg	Sandoglobulin	At the beginning of the 2nd dose	2.86	97% Granulocytes	Cefotaxime sodium 145 mg/kg/day for 72 h	[44]
Chronic inflammatory demyelinating polyradiculoneuro-pathy	F	62	0.4 g/kg	Ns	5th day	0.02	90% Granulocytes	Analgesics	[45]
ITP	F	44	0.6 g/kg	Gammagard	2nd day	1.83	83% granulocytes	Ceftriaxone 12 g/d	[46]
ITP	F	10	1 g/kg	Flebogamma	10 h after 2nd infusion	7.44	98% granulocytes	Cefotaxime 60 mg/kg every 6 h	[33]
ITP	F	6	1 g/kg	Flebogamma	12 h after 2nd infusion	0.65	60% granulocytes	Analgesics	[33]
Kawasaki syndrome	M	9	2 g/kg	Polygam	10 h after last infusion	1.515	99% granulocytes	Ceftriaxone for 72 h	[47]
Acquired immune neutropenia	Ns	2	1 g/kg	Sandoglobulin	During 2nd infusion	3.50	95% granulocytes	Self-limiting	[48]
ITP	M	7	0.4 g/kg	Sandoglobulin	12 h after 2nd infusion	1.620	95% granulocytes	Self-limiting	[49]
ITP	M	8	0.4 g/kg	Sandoglobulin	During 3rd infusion	0.667	92% granulocytes	Self-limiting	[49]
Systemic lupus with renal failure	F	42	2 g/kg	Octagam	2 days after infusion	2.710	94% granulocytes	Dexamethasone, vancomycin, meropenem	[50]
Kawasaki disease	F	6	1 g/kg	Sulfonated	Within 40 h of infusion	0.12	13% granulocytes	Methylprednisolone	[20]
Kawasaki disease	F	7	2 g/kg	Sulfonated	Within 25 h of infusion	0.648	83% granulocytes	Self-limiting	[20]
Kawasaki disease	F	10	1 g/kg	Peg-treated	Within 31 h of infusion	0.021	65% granulocytes	Self-limiting	[20]
Kawasaki disease	M	1	2 g/kg	Peg-treated	Within 33 h of infusion	1.248	89% granulocytes	Methylprednisolone	[20]
Common variable immunodeficiency	M	10	0.4 g/kg	NS	10 days after last infusion	0.225	87% lymphocytes	Ticarcillin-clavulanate and ofloxacin	[51]
Guillain-barre	M	14	0.4 g/kg	NS	4th day	0.0000018	85% lymphocytes	Hydration and analgesics	[34]
ITP	F	77	2 g/kg	Privigen	1st day	0.073	71% granulocytes	Antibiotics	[21]
ITP	F	35	1 g/kg	Privigen	1st day	0.476	66% lymphocytes	Ceftriaxone, vancomycin, ampicillin, acyclovir, analgesics	[21]
ITP	M	4	1 g/kg	Privigen	Within 2 h of infusion	0.393	42% granulocytes	Ciprofloxacin, vancomycin	[21]
Chronic inflammatory demyelinating polyneuropathy	M	49	2 g/kg	Gamunex	1 day after 3rd dose	NS	Small lymphocytes	Antibiotics	[21]
Warm autoimmune hemolytic anemia	M	20	1 g/kg	Gamunex	1st day	0.257	88% granulocytes	Ceftriaxone, acyclovir	[21]
ITP	F	80	1 g/kg	Privigen	1st day	NS	NS	Vancomycin, ceftriaxone, ampicillin	[21]
Primary immune deficiency	F	25	15 g	Gammagard liquid	3 days after last infusion	0.016	87% lymphocytes	Vancomycin, cefotaxime, analgesics, antiemetic	[21]
Myasthenia gravis	F	18	2 g/kg	Gamunex	Within 1–2 days of infusion	0.082	79% granulocytes	Vancomycin, chloramphenicol, analgesics, antiemetic	[21]
End-stage kidney disease	M	31	1 g/kg	NS	Less than 24 h after last infusion	3.846	90% granulocytes	Acyclovir, vancomycin, cefotaxime, amoxicillin	[52]
Systemic lupus erythematosus	F	46	2 g/kg	IVIG 10%	36 h after 1st infusion	1.547	87.5% granulocytes	Ceftriaxone 2 g every 12 h and ampicillin 2 g every 4 h	[53]
Acute EBV infection	M	4	0.4 g/kg	NS	6 h after 2nd infusion	2.993	84% granulocytes	Ceftriaxone, dexamethasone	[54]

NS—not stated, IVIG—intravenous immunoglobulins, CSF—cerebrospinal fluid, F—female, M—male, WBC—white blood cells, ITP—immune thrombocytopenic purpura.

**Table 2 jcm-11-03571-t002:** Overview of documented IVIG-related aseptic meningitis cases in the years 1981–2019.

Patient Cases	Age (mean)	Different Diagnosis	Brand Names	Gender Distribution	CSF Cytosis	WBC × 10^9^ L (CSF)	Antibiotic Therapy
44	22.4 years old	18	11 NS—14	M—22 F—21 NS—1	Granulocyte—32 Lymphocyte—8 NS—4	0.0000018–7.44	Yes—20 No—23 NS—1

NS—not stated.

## Data Availability

Not applicable.

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
