# Peer review of "Intravenous Immunoglobulin-Induced Aseptic Meningitis—A Narrative Review of the Diagnostic Process, Pathogenesis, Preventative Measures and Treatment"

_jcm, 2022, doi:10.3390/jcm11133571_

Round 1
Reviewer 1 Report
The manuscript attempted to review and analyze the intravenous immunoglobulin-induced aseptic meningitis. This review described 44 patient cases of IVIG-related aseptic meningitis with additional overview of the diagnosis, pathophysiology, possible preventative measures and treatment. However, the manuscript is not suitable for publication and needs to be substantially revised, because there are several limitations of this study, as shown below:
Major points:
1. The authors only made a brief summary of the independent cases published by others, but there was no systematic analysis in the manuscript, so this manuscript can not be called Meta-analysis, and the title is obviously inappropriate.
2. In addition, the authors only listed the cases published by others and did not summarize the cases data in detail. For example, Table 1 only listed the data, without basic analysis of corresponding characteristics (such as diagnosis, gender, age, IVIG dose, etc.). And the authors did not draw the conclusions of clinical significance.
Minor points
-The manuscript mainly discuss the diagnosis, pathophysiology, possible preventative measures and adequate treatment of aseptic meningitis. In Keywords, "immune thrombocytopenia."should be removed and replaced with other keywords.
-Line 65-66, the causative agents of drug induced aseptic meningitis (DIAM) have been thoroughly analyzed in literature. The author needs a brief analysis of this point.
-The Table 1 should be in three lines form.
Author Response
Dear Editor,
Thank you for the opportunity to respond to the Reviewers’ comments. We have followed all the suggestions we have received, thus, we believe we managed to improve our manuscript significantly. The additional time we received to edit the manuscript allowed us to precisely clarify the ambiguities and fill the gaps, therefore, contributing to our manuscript's improvement.
Response to Reviewer 1 Comments
Major Points
Point 1: The authors only made a brief summary of the independent cases published by others, but there was no systematic analysis in the manuscript, so this manuscript cannot be called Meta-analysis, and the title is obviously inappropriate.
Response 1: Taking into account the Reviewers’ comments, the title of the paper was changed to “Intravenous immunoglobulin-induced aseptic meningitis - a narrative review of the diagnostic process, pathogenesis, preventative measures and treatment”. The authors decided to alter the submitted article to fit the form of a narrative review type.
Point 2: In addition, the authors only listed the cases published by others and did not summarize the cases data in detail. For example, Table 1 only listed the data, without basic analysis of corresponding characteristics (such as diagnosis, gender, age, IVIG dose, etc.). And the authors did not draw the conclusions of clinical significance.
Response 2: The data included in Table 1 has now been analyzed under the section “Characteristics of IVIG-induced aseptic meningitis reported cases” and conclusions of clinical significance have been drawn.
“As of April 2022 to our knowledge, forty-four cases of intravenous immunoglobulin-associated aseptic meningitis have been reported in the English-language literature. The median age of the patients was 22,4 years, with twenty-four (54.50%) of the published cases involving children fourteen and younger. Therefore the incidence of this adverse reaction should be especially brought to the attention of pediatricians. IVIG-induced aseptic meningitis was diagnosed most often during the course of treatment of immune thrombocytopenic purpura (21 patients) and Kawasaki disease (5 patients). Noteworthy, roughly around 45% of the reported patient cases (20 patients) were treated with empiric antibiotic therapy regardless of the cerebrospinal fluid (CSF) analysis – 57.10% of patients with lymphocyte predominance in the CSF and only 46.80% of those with the majority of granulocytes in the CSF received antibiotic treatment (tabl.1). The onset of aseptic meningitis symptoms greatly varied depending on the patient, with the earliest side effects reported within 24-hours of the first infusion to even 10 days after the last one (22,50). Interestingly, white blood cell count in the cerebrospinal fluid of the described patients show a broad range from 0,0000018 x 10^9/L in a patient with aseptic meningitis in the course of treatment of Guillain-Barre syndrome to 7,44 x 10^9/L in a child with ITP (45,52). The gender distribution in the reported cases points to a slight dominance of males (22 males, 21 females, 1 not stated). Published literature on the subject of IVIG-induced aseptic meningitis refers to intravenous immunoglobulins under thirteen different brand names. (tabl.1) “
Minor points
Point 3: The manuscript mainly discuss the diagnosis, pathophysiology, possible preventative measures and adequate treatment of aseptic meningitis. In Keywords, "immune thrombocytopenia." should be removed and replaced with other keywords.
Response 3: Keywords have been changed to: “aseptic meningitis; intravenous immunoglobulins, drug-induced meningitis”. "Immune thrombocytopenia” was removed as suggested by the Reviewer.
Point 4: Line 65-66, the causative agents of drug induced aseptic meningitis (DIAM) have been thoroughly analyzed in literature. The author needs a brief analysis of this point.
Response 4: This point is analyzed later in the paragraph. This section provides a brief description of the most common causative agents of drug-induced aseptic meningitis, pointing to intravenous immunoglobulins (IVIGs) and non-steroidal anti-inflammatory drugs (NSAIDs) as the most common causes of DIAM. Other reasons for the incidence of drug-induced aseptic meningitis include treatment with cotrimoxazole, cephalosporins, intrathecal methylprednisolone and anesthetics. The aim of this paragraph was to identify IVIG as one of the most common causes of DIAM. The pathophysiology of DIAM is later thoroughly analyzed under the heading “Pathophysiology of IVIG-associated aseptic meningitis”.
Point 5: The Table 1 should be in three lines form.
Response 5: The table was changed into three lines form.
Reviewer 2 Report
Surprisingly little has been published about IVIG induced aseptic meningitis and I thank the authors for their valuable work. However, I have serious concerns about the article in its current form.
major
- the title suggests a systematic review and meta-analysis. However, the current form seems more like a narrative review to me. The authors do not specify their search (I can’t redo the search with the current information), quality of the articles are not described, and apart from table 1 very few aggregated results are presented and metaregression is also not done/possible.
- the PRISMA checklist is more than a format for the flow chart of study identification and selection. A helpful next step could be to answer all the items.
- If a more numeric systematic review is what the authors want to do, then I would suggest to limit the question the authors want to answer. For instance incidence, dose-effect, presenting symptoms, or outcome. If a more general article about this topic is aimed for, then I would suggest to use the identified articles to write a narrative review. This is in line with what the authors have written so far: “an overview of diagnosis, pathophysiology, possible preventative measures and adequate treatment of aseptic meningitis.” But I would then narrow it to IVIG induced aseptic meningitis.
- Section 4 “Diagnosis of aseptic meningitis”. This is an example of what I am trying to convey with my previous remarks. What is the aim of this section? To describe how aseptic meningitis should be diagnosed in clinical practice? To describe the different definitions of aseptic meningitis that are used in the included articles in the review? To determine in advance what the authors think that the definition is and use their definition to select the articles included in this review? All are fine, but it should be clear.
- In the interpretation of the results a more thorough discussion could be provided about the (un)certainty about the causal relationship between IVIG and meningitis. How common is aseptic meningitis in the most common diseases for which IVIG is prescribed when IVIG is not (or not yet) started?
Author Response
Dear Editor,
Thank you for the opportunity to respond to the Reviewers’ comments. We have followed all the suggestions we have received, thus, we believe we managed to improve our manuscript significantly. The additional time we received to edit the manuscript allowed us to precisely clarify the ambiguities and fill the gaps, therefore, contributing to our manuscript's improvement.
Response to Reviewer 2 Comments
Point 1: the title suggests a systematic review and meta-analysis. However, the current form seems more like a narrative review to me. The authors do not specify their search (I can’t redo the search with the current information), quality of the articles are not described, and apart from table 1 very few aggregated results are presented and metaregression is also not done/possible.
Response 1: Taking into account the reviewers’ comments, the title of the paper was changed to “Intravenous immunoglobulin-induced aseptic meningitis - a narrative review of the diagnostic process, pathogenesis, preventative measures and treatment”. The authors decided to alter the submitted article to fit the form of a narrative review type.
Point 2: the PRISMA checklist is more than a format for the flow chart of study identification and selection. A helpful next step could be to answer all the items.
Response 2: Several more detailed steps have been described in the materials and methods section in order to provide for a more thorough analysis of publication selection.
"A comprehensive search was conducted last in April 2022 in PubMed online electronic database using the key phrases (“immunoglobulin” OR “IVIG”) AND “aseptic meningitis”. No timeframe restrictions were assigned for the selected publications. The articles were selected following the PRISMA guidelines. A total of 284 articles were found. Before the initial screening 3 articles were found to have duplicates, therefore they were removed. Additionally 31 papers were unable to be screened (i.e. no abstract available). 250 publications were screened and 22 non-English articles were excluded. 5 articles were unable to be retrieved. Out to the 223 reports, which were assessed for eligibility, 21 were rejected as they involved non-human subjects. 128 articles were identified after thorough analysis as irrelevant to the review. Finally, a total of 74 papers were analyzed in this article. In Fig. 1, we present the schematic diagram of the selection process of articles chosen for this review."
Point 3: If a more numeric systematic review is what the authors want to do, then I would suggest to limit the question the authors want to answer. For instance incidence, dose-effect, presenting symptoms, or outcome. If a more general article about this topic is aimed for, then I would suggest to use the identified articles to write a narrative review. This is in line with what the authors have written so far: “an overview of diagnosis, pathophysiology, possible preventative measures and adequate treatment of aseptic meningitis.” But I would then narrow it to IVIG induced aseptic meningitis.
Response 3: The goal of the submitted article was to provide a general but thorough review on the concept of IVIG-induced aseptic meningitis, give an in-depth analysis of the way of clinically diagnosing AM, the pathogenesis, preventative measures and treatment of the disease. The aim was to write a comprehensive review, especially beneficial to praciticing clinicians, allowing for easier and quicker diagnosis of AM.
Point 4: Section 4 “Diagnosis of aseptic meningitis”. This is an example of what I am trying to convey with my previous remarks. What is the aim of this section? To describe how aseptic meningitis should be diagnosed in clinical practice? To describe the different definitions of aseptic meningitis that are used in the included articles in the review? To determine in advance what the authors think that the definition is and use their definition to select the articles included in this review? All are fine, but it should be clear.
Response 4: This section provides an in-depth description of a clinical approach to diagnosing IVIG-related aseptic meningitis; therefore the heading of the section was changed to make this clear. The aim of this part of the article was to underline the possible clinical manifestations of aseptic meningitis and the possible measures of diagnosis based on the thorough analysis of reported patient cases in order to provide a comprehensive overview of the subject.
Point 5: In the interpretation of the results a more thorough discussion could be provided about the (un)certainty about the causal relationship between IVIG and meningitis. How common is aseptic meningitis in the most common diseases for which IVIG is prescribed when IVIG is not (or not yet) started?
Response 5: Based on the literature review, most common diseases for which IVIG is prescribed include immune thrombocytopenia purpura and Kawasaki disease. A thorough search was performed to provide response to these very interesting questions of the reviewer. The authors found “very few associations in literature to have been made between the diagnosis of ITP and the incidence of aseptic meningitis, showing a direct cause-and-effect relationship between intravenous immunoglobulin treatment and occurrence of aseptic meningitis. A study by Mohammed A. Aldriweesh et al. identifies Varicella Zoster Virus as both the most common viral infection among patients with ITP and one of the leading viral causes of aseptic meningitis. The possibility of a concomitant incidence of both ITP and AM was also reported in the context of a potential adverse reaction of measles-mumps containing vaccination by Silvia Perez-Vilar et al. On the contrary, patients with Kawasaki disease may manifest in other, less common ways such as aseptic meningitis. Therefore the incidence of AM in the course of Kawasaki disease therapy may not always be directly associated with intravenous immunoglobulin infusion. “
Round 2
Reviewer 1 Report
The authors have substantially revised this manuscript according to the comments. I am basically satisfied with the revised version, but I suggest adding a new Table to explain "3. characteristics of IVIG induced aseptic meningitis reported cases" to make it more clear.
Author Response
Dear Editor,
Dear Reviewer,
Thank you for the opportunity to respond to the Reviewers’ comments. We have followed all the suggestions we have received, thus, we believe we managed to improve our manuscript significantly. The article has been spell checked and minor changes were made.
Response to Reviewer 1 Comments
Minor Points
Point 1:The authors have substantially revised this manuscript according to the comments. I am basically satisfied with the revised version, but I suggest adding a new Table to explain "3. characteristics of IVIG induced aseptic meningitis reported cases" to make it more clear.
Response 1: A new table (Table 1) was added in order to better visualize section 3 of the article-"characteristics of IVIG induced aseptic meningitis reported cases".
Reviewer 2 Report
I do not have many additional comments as most of my previous points have been addressed. I think my point about the prisma checklist was misunderstood. I meant that the PRISMA checklist (e.g. http://www.prisma-statement.org/documents/PRISMA_2020_checklist.pdf ) should/could be used throughout the whole manuscript for structure and completeness, not just the identification of studies to be included. The authors/editors could consider checking if all relevant items are included by completing the checklist and uploading it as supplementary material. However, this approach may be less essential for a narrative review.
Author Response
Dear Editor,
Dear Reviewer,
Thank you for the opportunity to respond to the Reviewers’ comments. We have followed all the suggestions we have received, thus, we believe we managed to improve our manuscript significantly. The article has been spell checked and minor changes were made.
Response to Reviewer 1 Comments
Minor Points
Point 1:I do not have many additional comments as most of my previous points have been addressed. I think my point about the prisma checklist was misunderstood. I meant that the PRISMA checklist (e.g. http://www.prisma-statement.org/documents/PRISMA_2020_checklist.pdf ) should/could be used throughout the whole manuscript for structure and completeness, not just the identification of studies to be included. The authors/editors could consider checking if all relevant items are included by completing the checklist and uploading it as supplementary material. However, this approach may be less essential for a narrative review.
Response 1: A PRISMA checklist is included in the supplementary material.